# Life Cycle Sustainability Performance Assessment Method for Comparison of Civil Engineering Works Design Concepts: Case Study of a Bridge

**DOI:** 10.3390/ijerph17217909

**Published:** 2020-10-28

**Authors:** Kristine Ek, Alexandre Mathern, Rasmus Rempling, Petra Brinkhoff, Mats Karlsson, Malin Norin

**Affiliations:** 1NCC AB, Gullbergs Strandgata 2, 405 14 Göteborg, Sweden; alexandre.mathern@chalmers.se (A.M.); rasmus.rempling@chalmers.se (R.R.); petra.brinkhoff@ncc.se (P.B.); malin.norin@ncc.se (M.N.); 2Department of Architecture and Civil Engineering, Chalmers University of Technology, 6, 412 96 Göteborg, Sweden; mats.d.karlsson@trafikverket.se; 3Swedish Transport Administration, Bataljonsgatan 8, 553 05 Jönköping, Sweden

**Keywords:** sustainability, life cycle assessment, life cycle costing, environmental externalities, indicator, multi-criteria decision analysis, civil engineering, bridge, design

## Abstract

Standardized and transparent life cycle sustainability performance assessment methods are essential for improving the sustainability of civil engineering works. The purpose of this paper is to demonstrate the potential of using a life cycle sustainability assessment method in a road bridge case study. The method is in line with requirements of relevant standards, uses life cycle assessment, life cycle costs and incomes, and environmental externalities, and applies normalization and weighting of indicators. The case study involves a short-span bridge in a design-build infrastructure project, which was selected for its generality. Two bridge design concepts are assessed and compared: a concrete slab frame bridge and a soil-steel composite bridge. Data available in the contractor’s tender phase are used. The two primary aims of this study are (1) to analyse the practical application potential of the method in carrying out transparent sustainability assessments of design concepts in the early planning and design stages, and (2) to examine the results obtained in the case study to identify indicators in different life cycle stages and elements of the civil engineering works project with the largest impacts on sustainability. The results show that the method facilitates comparisons of the life cycle sustainability performance of design concepts at the indicator and construction element levels, enabling better-informed and more impartial design decisions to be made.

## 1. Introduction

In civil engineering projects, life cycle environmental, social, and economic sustainability performance is becoming increasingly important, as reflected in the large number of standards published on the subject in recent years [1,2,3,4]. To make better-informed decisions regarding the impact of design choices on the sustainability of civil engineering works, sustainability performance assessment is recommended [5,6]. It is important that assessments are performed in a harmonized way and can be compared impartially. Current standards provide the general framework for the sustainability assessment of civil engineering works but do not give detailed guidance on the calculation of indicators and their aggregation [2,4]. In most studies on sustainability-based design and optimization of bridges, simplifications are used, and the assessment is based on one or two selected indicators and only covers certain life cycle stages [6], e.g., CO_2_ emissions and the cost of construction materials [7] and of transport and installation [8] and embodied energy of construction materials [9].

The potential to influence the sustainability of a design is larger in the early stages of the design process than in later stages [10]. It is therefore important to define indicators that can support an iterative sustainability-driven design process from concept to final implementation. To enable the identification of sustainable designs, a formalized method that allows transparent, comparable, and automatable sustainability design and assessment is desired.

Ek et al. presented a harmonized method for life cycle sustainability assessment and comparison of civil engineering works design concepts [11]. The proposed method includes guidance on the calculation of environmental, social and economic indicators, based on life cycle assessment (LCA), life cycle costing (LCC) and external costs, and aggregation using normalization and weighting factors, in accordance with the principles and requirements of methods for sustainability performance assessment given in the standards [2] and [4]. 

This paper evaluates the previously proposed method by applying it in a road bridge case study. The study has two primary aims: (1) to analyse the practical application potential of the method in carrying out transparent sustainability assessments of design concepts in the early planning and design stages and (2) to examine the results obtained in the case study to identify critical indicators in different life cycle stages as well as critical elements in the civil engineering works project with the greatest impacts.

Life cycle stages are classified into so-called modules by the related standards [1,2,3,4,12,13,14,15], see Figure 1.

Module A0 is the pre-construction stage. Modules A1–A3 represent the production stage from raw material extraction to construction material manufacturing where A1 is material extraction, A2 is transport from extraction to manufacture, and A3 is material manufacture. A4–A5 represent the construction process stage where A4 is transport from material manufacture to the construction site and A5 is the construction site works. B1–B5 represent the use stage relating to maintenance where B1 is normal use of the bridge, B2 is maintenance, B3 is repair, B4 is replacement, and B5 is refurbishment. B6–B7 represent the use stage related to the operation where B6 is operational energy use and B7 is operational water use. B8 is the use stage related to the user’s utilization of the civil engineering works. C1–C4 cover the end-of-life stage where C1 is deconstruction, C2 is transport from the deconstruction site to the waste management site, C3 is the waste processing of materials intended for reuse, recycling, and energy recovery, and C4 is waste disposal. Module D represents the benefits and loads beyond the system boundary of the civil engineering works.

## 2. Materials and Methods

The new method for life cycle sustainability assessment and comparison of civil engineering works design concepts presented by Ek et al. [11] was applied to a case study. The case study involved a bridge in a design-build infrastructure project and was selected for its generality. Two alternative bridge design concepts were assessed and compared: a concrete slab frame bridge (CSF bridge) and a soil-steel composite bridge (SSC bridge). The prerequisites for the assessment are presented in Table 1.

The functional unit is 1 m of bridge length and year of required service life (RSL). The reference study period is equal to the required service life: 80 years. A functional equivalent of 1 km of bridge and year of RSL is prescribed by the product category rules (PCR) of the International EPD System [13], but it was decided to use 1 m instead, because the short length of the bridge (6 m) would yield non-representative results if scaled to 1 km.

Life cycle assessment (LCA) was performed according to the standard EN 15804 [12] using the LCA software GaBi Professional, version 9.5 (Sphera Solutions GmbH, Leinfelden-Echterdingen, Germany) [16]. The GaBi datasets used are presented in Appendix A. Normalization and weighting factors presented in [11] (adapted from the factors used in the Product Environmental Footprint (PEF) method [17,18,19] were used (see Table 2). As proposed in [11], some indicators are categorized into the environmental dimension and some into the social dimension. In PEF, all indicators are in a single dimension. The PEF weighting factors of the indicators have thus been scaled to a total of 100 in the environmental and social dimension, respectively. The life cycle costing (LCC and incomes) was calculated according to the standard EN 15686-5 [14], and environmental externalities were calculated in accordance with ISO 14008 [15]. The economic indicators are presented separately in line with the standards’ requirements. LCC and incomes as well as environmental externalities are presented as the net present value (NPV) using a discount rate of 3%. This discount rate was chosen as it is the rate prescribed by the currently available standard on calculation methods for economic performance (for buildings) [20]. Environmental externalities were calculated using the EPS 2015dx method [21]. Abbreviations and units of measurement used for the indicators included in the assessment are presented in Table 3.

The method requires the use of a life cycle inventory (LCI) to calculate the indicators. In the case study, the LCI was in the form of a bill of materials (BOM), which was calculated for each design concept for modules A1–A5 as well as for the use (B1–B8) and end-of-life (C1–C4 and D) stages. It was created based on data available in the tender phase. The BOM is presented in Appendix B. The values of modules B1–B8, C1–C4, and D were calculated based on the scenarios presented in Appendix C. Realistic and representative scenarios of the resource consumption and costs of modules B–D were developed for each design concept. Scenarios were developed based on project documentation and literature data, if available, as well as expert knowledge. Expert knowledge was based on both the project manager of the construction project used as case study, and the authors’ previous experience from bridge design, construction and maintenance projects in Sweden. 

For the LCC, the average market prices per unit for each of the resources included were used. Average prices were supplied by the project manager of the case study project. 

Transport modes and distances are presented in Appendix D. Cut-offs were made for the transport of form oil, bitumen sheet, bitumen sealant, impregnation, geotextile, mortar, graffiti protection, polypropylene pipe, polyethylene foam, bituprimer, epoxy sealant, and plastic film.

## 3. Results

The results are first presented separately for the CSF bridge design concept and the SSC bridge design concept. The results for the two concepts are then compared. Positive economic indicator values indicate costs, while negative values indicate incomes. For the environmental and social indicators, positive values indicate a negative impact, and negative values indicate a positive impact. 

The result for module B8 is presented in a separate figure, since it is relatively higher than that of the other modules. Furthermore, the result for module B8 was excluded from the comparison of the concepts, since it would have disguised the differences in concepts among the other modules. In addition, the result for module B8 was found to be equal for both concepts.

### 3.1. Concrete Slab Frame Bridge Design Concept

The results for each indicator of the sustainability assessment of the CSF bridge in units of measurement and per life cycle stage are presented in Table 4.

The results for the CSF bridge are presented per life cycle stage in Table 5. The results for the environmental and social dimensions are aggregated on the dimension level, using the normalization and weighting factors in Table 2, while the results for the economic dimension are summarized at the indicator level. The normalized and weighted results for the environmental and social dimensions are presented per life cycle stage and per indicator in Figure 2 (excluding module B8). Figure 3 shows the contribution of each resource to the total impact over the life cycle (modules A–C excluding B8) in the environmental dimension and the social dimension, respectively, for the resources with the greatest contributions. The normalized and weighted results for the environmental and social dimensions are presented for module B8 per indicator in Figure 4.

In the environmental dimension, 52% of the total impact of life cycle stages A–C, excluding module B8, occurs in the production stage (modules A1–A3), followed by the construction stage (36%, modules A4–A5). If included, B8 would contribute to 50% of the total impact for life cycle stages A–C. Module D presents the potential to reduce the environmental impact of life cycle stages A–C (excluding module B8) by 35% through future re-use or recycling. The main contribution to the environmental impact over the life cycle (modules A–C, excluding B8) comes from the indicators “global warming potential” (44% of the total impact) and “abiotic depletion potential for fossil resources” (28% of the total impact, see Figure 2). Fifty-one percent of the global warming potential is caused by the production (29%) and transport to the construction site (22%) of 6833 tons of aggregates, while 23% is caused by the production of 472 tons of concrete, and 6% is caused by the production and combustion of diesel used on the site during construction. The production and transport of aggregates and the landfilling of aggregate waste contributes to 53% of the total environmental impact over the life cycle (see Figure 3).

In the social dimension, 67% of the total impact for life cycle stages A–C (excluding module B8) occurs in the production stage, followed by the construction stage (24%). If included, module B8 would contribute to 66% of the total impact for life cycle stages A–C. Module D presents the potential to reduce the social impact by 33% of the total impact for life cycle stages A–C (excluding module B8) by future re-use or recycling. The main contribution to social impact over the life cycle (modules A–C, excluding B8) comes from the indicators “particulate matter emissions” (49% of the total impact) and “human toxicity—cancer effects” (27% of the total impact). Thirty-five percent of the particulate matter emissions are caused by the production of aggregates and their transport to the construction site, and 28% is caused by the production and combustion of diesel at the construction site during construction. The production and transport of aggregates and the landfilling of aggregate waste contributes to 38% and reinforcement steel and steel rack production contributes to 29% of the total social impact over the life cycle (see Figure 3).

In the economic dimension, 54% of the total net cost for the indicator “LCC and incomes” in life cycle stages A–C (excluding module B8) occurs in the construction stage (mainly in A5), see Table 5. Forty percent occurs in the production stage. If module B8 was included, it would contribute to 14% of the total net cost for life cycle stages A–C. Module D presents the potential to reduce the future net cost (by future re-use or recycling) by 0.2% for life cycle stages A–C (excluding module B8). The main contributor (30%) to the cost of the LCC and incomes over the life cycle is the work costs during construction, followed by the costs for production and transport of aggregates (18%) and for the production and transport of concrete and reinforcement steel (17%). For the indicator “environmental externalities”, 68% of the total cost in life cycle stages A–C (excluding module B8) occurs in the production stage, followed by the construction stage (31%). If module B8 was included, it would contribute to 12% of the total cost for life cycle stages A–C. Module D presents the potential to reduce future costs of the total impact for life cycle stages A–C (excluding module B8) by 4%. The main contributors (43%) to the environmental externalities over the life cycle are the production and transport of aggregates and transport to waste disposal and the landfilling of aggregate waste. 

#### 3.1.1. Modules A1–A3

In the production stage, greenhouse gas emissions have the largest impact in the environmental dimension (see Figure 2). The indicator “global warming potential” contributes to 49% of the total impact in this life cycle stage. Fifty percent of this originates from the production of 6833 tons of aggregates, 39% from the production of 472 tons of concrete, and 9% from the production of 23 tons of reinforcement steel (carbon and stainless). The production of asphalt, steel racks, and other built-in construction materials accounts for 2% of the global warming potential.

The second largest impact comes from the abiotic depletion potential for fossil resources; this constitutes 29% of the total environmental impact. Sixty-eight percent of this depletion is caused by the production of aggregates, 16% by the production of concrete, and 11% by the production of reinforcement steel.

The “acidification potential”, the “eco-toxicity potential”, and the “photochemical ozone creation potential” indicators also have significant impacts, each accounting for about 5% of the total environmental impact. The production of aggregates contributes to 46–69% of the total impact for each of these indicators.

The largest impact in the social dimension comes from the indicators “human toxicity—cancer effects” (38%) and “particulate matter emissions” (36%). For “human toxicity—cancer effects”, 97% of the impact comes from the production of 553 kg of stainless-steel reinforcement. For “particulate matter emissions”, 66% of the impact comes from the production of aggregates.

In the economic dimension, for the indicator “LCC and incomes”, 39% of the total cost is for the aggregates, 23% is for the ready-mix concrete, and 19% is for carbon steel reinforcement. There is no income. For the indicator “environmental externalities”, 36% of the total cost is for the production of aggregates, and 20% each is for the production of stainless-steel reinforcement and ready-mix concrete.

#### 3.1.2. Modules A4–A5

In the construction process stage, greenhouse gas emissions have the largest impact in the environmental dimension (see Figure 2). The global warming potential accounts for 39% of the total impact in this life cycle stage. Seventy percent of this type of emission originates from the transport of aggregates from the production site to the construction site. Eighteen percent originates from the production and combustion of diesel in construction machines on the construction site.

The second largest impact comes from the abiotic depletion potential for fossil resources, having 26% of the total environmental impact. Sixty-eight percent of this is caused by the transport of aggregates to the site.

The third largest impact comes from the combination of the three indicators for eutrophication potential, together contributing to 9% of the total environmental impact. Approximately 72% of the eutrophication potential (marine) and eutrophication potential (terrestrial) comes from the production and combustion of diesel used on site, while 65% of the eutrophication potential (freshwater) comes from the transport of aggregates.

Significant impacts are also caused by the “eco-toxicity potential”, the “potential soil quality index”, and the “photochemical ozone creation potential” indicators, each accounting for 6–7% of the total environmental impact. The transport of aggregates contributes to 69% of the eco-toxicity potential. Ninety percent of the potential soil quality index value comes from the production of the ancillary materials wood, particleboard, and plywood for the formworks. The production and combustion of diesel used on the construction site contributes to 74% of the photochemical ozone formation potential.

The largest impact in the social dimension comes from particulate matter emissions (84%). Sixty-nine of this is caused by the production and combustion of diesel used on the construction site.

In the economic dimension, for the indicator “LCC and incomes”, 65% of the total cost is for the construction workers and 11% is for the transport of aggregates. There is no income. For the indicator “environmental externalities”, 62% of the total cost is for the transport of aggregates, and 25% is for the production and combustion of diesel used on the construction site.

#### 3.1.3. Modules B1–B7

In the use stage, “global warming potential” (38%) has the largest impact in the environmental dimension, followed by “abiotic depletion for fossil resources” (35%), see Figure 2. Fifty-three percent of the global warming potential is caused by the production of 1.4 tons of steel racks, and 30% is caused by the production of 14 tons of ready-mix concrete. Thirty-two percent of the abiotic depletion potential for fossil resources is caused by the production of 10.5 tons of asphalt. 

“Particulate matter emissions” (58% of the total impact) has the largest impact in the social dimension, followed the indicator “human toxicity—non-cancer effects” (23%). Thirty-nine percent of the particulate matter emissions is caused by the production of steel racks, and 21% is caused by the production of 35 kg of bituprimer. The largest contributor to “human toxicity—non-cancer effects” is the production of 77 kg of epoxy (45%). 

In the economic dimension, for the indicator “LCC and incomes”, 46% of the total cost is for steel racks. The cost for installation of the steel racks is 21% of the total cost. There is no income. For the indicator “environmental externalities”, 61% of the total cost is for the production of steel racks, and 23% is for the production of asphalt. 

#### 3.1.4. Module B8

The impact of module B8, the stage relating to the user’s utilization, is the same for both design concepts (see Figure 4, Table 5 and Table 6). For the CSF bridge, the environmental impact of module B8 is equal to the environmental impact of all other modules together (excluding module D). The social impact of module B8 is 97% greater. The net cost for LCC and incomes is 84% lower for module B8 than for all other modules together (excluding module D). The “environmental externalities” indicator is 83% lower for module B8 than for all other modules together (excluding module D). 

The indicators “abiotic depletion potential for fossil resources” and “global warming potential” account for 21% and 19% of the total impact, respectively, in the environmental dimension. Fifteen percent comes from the indicator “acidification potential”, and 13% comes from the indicator “eutrophication potential” (terrestrial). However, if the results for all three eutrophication indicators are pooled, they account for 23% of the total environmental impact. Fifty-four percent of the global warming potential originates from the production and combustion of hydrogenated vegetable oil (HVO) (rapeseed methyl ester (RME) was used as a proxy for HVO). Fifty-eight percent of the abiotic depletion potential for fossil resources originates from the production of electricity. Seventy-six percent of the total eutrophication potential originates from the production and combustion of HVO.

Forty-seven percent of the total impact in the social dimension comes from the indicator “human toxicity potential—non-cancer effects”, and 34% comes from “particulate matter emissions”. Ninety-seven percent of the “human toxicity potential—non-cancer effects” originates from the production of HVO. For particulate matter emissions, 48% originates from the production of HVO, and 45% comes from the combustion of diesel.

In the economic dimension, for the indicator “LCC and incomes”, 46% of the total cost is for steel racks. The cost for installation of the steel racks is 21% of the total cost. There is no income. For the indicator “environmental externalities”, the largest contributor is the production of HVO. 

#### 3.1.5. Modules C1–C4

In the end-of-life stage, global warming potential has the largest impact in the environmental dimension (38%), followed by abiotic depletion potential for fossil resources (23%), the three eutrophication potential indicators together (12%), and photochemical ozone creation potential (11%), see Figure 2. Forty-six percent of the global warming potential and 49% of the abiotic depletion potential for fossil resources are caused by the landfilling of 683 tons of aggregate waste. Nineteen percent of the global warming potential and 21% of the abiotic depletion potential for fossil resources is caused by the production and combustion of diesel used in deconstruction.

Particulate matter emissions have the largest impact on the social dimension (56% of the total impact), followed by the indicator “human toxicity—non-cancer effects” (35%). Thirty-two percent of the particulate matter emissions is caused by the transport of 683 tons of aggregate waste, and 23% is caused by the production and combustion of diesel used during deconstruction of the bridge. Ninety percent of the impact of “human toxicity—non-cancer effects” is caused by the landfilling of aggregate waste.

In the economic dimension, for the indicator LCC and incomes, 93% of the total cost is for deconstruction workers. There is no income. For the indicator “environmental externalities”, the largest part of the cost (52%) comes from the landfilling of aggregate waste and the second largest (20%) portion comes from the production and combustion of diesel used for deconstruction.

#### 3.1.6. Module D

Regarding benefits and loads beyond the system boundary, the largest benefit in the environmental dimension is offered by the potential avoidance of contributing to global warming potential (45% of the total benefit) and the potential avoidance of contributing to the abiotic depletion potential for fossil resources (34%), see Figure 2. The main part of this benefit is due to the potential re-use of aggregates.

The largest benefit in the social dimension is offered by the potential avoidance of particulate matter emissions (55%) and the avoidance of ionizing radiation (21%). Here, as well, the main part of this benefit is due to the potential re-use of aggregates.

In the economic dimension, for the indicator “LCC and incomes”, 41% of the total income comes from the potential recycling of concrete as a filling material, and 37% comes from the potential recycling of carbon steel reinforcement. For the indicator “environmental externalities”, the largest benefit comes from the potential re-use of aggregates (48%).

### 3.2. Soil-Steel Composite Bridge Design Concept

The results for each indicator used in the sustainability assessment of the SSC bridge per metre of bridge in units of measurement and per life cycle stage are presented in Table 7.

The results for the SSC bridge are presented per life cycle stage in Table 6. The results for the environmental and social dimensions are aggregated on the dimension level, while the results for the economic dimension are aggregated on the indicator level. The results for the environmental and social dimensions are normalized and weighted using the factors in Table 2, and the results for the economic indicators are summarized. Normalized and weighted results for the environmental and social dimensions per life cycle stage are presented per indicator in Figure 4 (only module B8) and Figure 5 (excluding module B8). Figure 6 shows the share of contribution of each resource to the total impact over the life cycle (modules A–C excluding B8) in the environmental dimension and the social dimension respectively, for the resources with the greatest contributions.

In the environmental dimension, 72% of the total impact for life cycle stages A–C (excluding module B8) occurs in the production stage (modules A1–A3), followed by the construction stage (20%, modules A4–A5). If module B8 was included, it would contribute to 38% of the total impact for life cycle stages A–C. Module D has the potential to reduce the future environmental impact (by future re-use or recycling) by 31% for life cycle stages A–C (excluding module B8). The main contribution to the environmental impact over the life cycle (modules A–C, excluding B8) is accounted for by the indicators “abiotic depletion potential for non-fossil resources” (35% of the total impact), “global warming potential” (30% of the total impact), and “abiotic depletion potential for fossil resources” (19% of the total impact). Practically all (99.6%) of the abiotic depletion potential for non-fossil resources and 36% of the global warming potential are caused by the production of structural steel for the bridge. Forty-seven percent of the global warming potential is caused by the production and transport of aggregates. The production of structural steel for the bridge contributes to 54% and the production and transport of aggregates contributes to 30% of the total environmental impact over the life cycle (see Figure 6).

In the social dimension, 64% of the total impact of life cycle stages A–C (excluding module B8) occurs in the production stage, followed by the construction stage (25%). If module B8 was included, it would contribute to 73% of the total impact for life cycle stages A–C. Module D presents the potential to reduce the future social impact by 56% for life cycle stages A–C (excluding module B8). The main contributor to the social impact over the life cycle (modules A–C excluding B8) is accounted for by the indicators “particulate matter emissions” (66% of the total impact) and “human toxicity—non-cancer effects” (11% of the total impact). The largest portion of the particulate matter emissions is caused equally by the production of aggregates and the production and combustion of diesel used for the construction, maintenance, and deconstruction of the bridge over the life cycle (each 36%), followed by the production of structural steel (23%). Over the life cycle, the production and transport of aggregates contributes to 44%, the production of structural steel for the bridge contributes to 26% and the production and combustion of diesel used for the construction, maintenance, and deconstruction of the bridge over the life cycle contributes to 21% of the total social impact (see Figure 6).

In the economic dimension, 70% of the total net cost for the indicator “LCC and incomes” in life cycle in stages A–C (excluding module B8) occurs in the production stage, and 24% occurs in the construction stage (see Table 6). If module B8 was included, it would contribute to 17% of the total net cost for life cycle stages A–C. Module D presents the potential to reduce the future net cost (by future re-use or recycling) by 0.2% for life cycle stages A–C (excluding module B8). The greatest contributor (39%) to the cost of the LCC and incomes over the life cycle is the production of structural steel for the bridge. For the indicator environmental externalities, 68% of the total external cost in life cycle stages A–C (excluding module B8) occurs in the production stage. Ninety percent of the environmental externalities are caused by the production of structural steel for the bridge. If module B8 was included, it would contribute to 3% of the total external costs for life cycle stages A–C. Module D presents the potential to reduce the future external costs by 1% for life cycle stages A–C (excluding module B8). 

#### 3.2.1. Modules A1–A3

In the production stage, the abiotic depletion potential for non-fossil resources has the largest impact in the environmental dimension (see Figure 5). This indicator has 48% of the total impact in this life cycle stage. Almost all (99.7%) of the depletion is caused by the production of 40 tons of structural steel. The second largest contributor is the global warming potential; this constitutes 26% of the total environmental impact. Fifty-seven percent of this indicator is accounted for by the production of structural steel plates for the bridge and 42% by the production of aggregates. The abiotic depletion potential for fossil resources also has a significant impact (16% of the total environmental impact). 

Particulate matter emissions have the largest impact in the social dimension (60%). Fifty-eight of this originates from the production of aggregates and 40% comes from the production of bridge steel.

In the economic dimension, for the indicator “LCC and incomes”, 67% of the total cost is for structural steel and 28% is for aggregates. There is no income. For the indicator “environmental externalities”, 96% of the cost is for the production of structural steel.

#### 3.2.2. Modules A4–A5

In the construction stage, greenhouse gas emissions have the largest impact in the environmental dimension (see Figure 5). The global warming potential contributes to 41% of the total impact in this life cycle stage. Seventy-three percent of this originates from the transport of aggregates to the construction site. Nineteen percent originates from the production and combustion of diesel used in construction machines on the construction site. The second largest impact comes from the abiotic depletion potential for fossil resources, causing 27% of the total environmental impact. Seventy-two percent of this is caused by the transport of aggregates to the site. The three eutrophication potential indicators make a significant contribution, together contributing to 9% of the total environmental impact. The production and combustion of diesel used at the construction site contributes to 71% of the total eutrophication potential. 

In the social dimension, particulate matter emissions have the largest impact by far (86%). Eighty-seven percent of these emissions are caused by the production and combustion of diesel used at the construction site.

In the economic dimension, for the indicator “LCC and incomes”, 32% of the total cost is for the transport of aggregates, 28% is for the construction workers, and 20% is for transport of the bridge’s structural steel plates. There is no income. For the indicator “environmental externalities”, 66% of the total cost is for the transport of aggregates and 27% is for the production and combustion of diesel used at the construction site.

#### 3.2.3. Modules B1–B7

In the use stage, greenhouse gas emissions have the largest impact in the environmental dimension (42%), followed by the depletion of fossil resources (33%) (see Figure 5). Seventy-three percent of the global warming potential is caused by the production of 1.4 tons of steel racks and 16% is caused by the production of 10.5 tons of asphalt. Fifty percent of the abiotic depletion potential of fossil resources is caused by the production of asphalt and 41% is caused by the production of steel racks.

In the social dimension, particulate matter emissions have the greatest impact (59% of the total impact), followed by 16% each from the indicators “human toxicity—cancer effects” and “human toxicity—non-cancer effects”. Seventy-six percent of particulate matter emissions comes from the production of steel racks. The production of steel racks contributes to 87% of the factor “human toxicity—cancer effects” and 55% of “human toxicity—non-cancer effects”. The production of asphalt contributes to 31% of “human toxicity—non-cancer effects”.

In the economic dimension, for the indicator “LCC and incomes”, 50% of the total cost is for steel racks. The installation of steel racks accounts for 23% of the total cost. There is no income. For the indicator “environmental externalities”, 81% of the total cost is for the production of steel racks. 

#### 3.2.4. Module B8

The environmental impact of module B8, the stage relating to the user’s utilization, is 38% lower than the environmental impact of all other modules together (excluding module D) for the SSC bridge, see Table 6. The social impact of module B8 is almost three times larger. The net cost for LCC and incomes is 80% lower for module B8 than for all other modules together (excluding module D). The “environmental externalities” indicator is 97% lower for module B8 than for all other modules together (excluding module D). For other aspects of module B8 that do not depend on the bridge type, see Section 3.1.4.

#### 3.2.5. Modules C1–C4

In the end-of-life stage, global warming potential has the largest impact in the environmental dimension (34%), followed by the abiotic depletion potential for fossil resources (22%), the three eutrophication potential indicators together (14%), and the photochemical ozone creation potential (12%), see Figure 5. Fifty-one percent of both the global warming potential and the abiotic depletion potential for fossil resources is caused by the landfilling of 683 tons of aggregate waste. Thirty-three percent of the global warming potential is caused by the production and combustion of diesel used for deconstruction. Thirty-four percent of the abiotic depletion potential for fossil resources is caused by the production of diesel used for deconstruction.

In the social dimension, particulate matter emissions have the greatest impact (56% of the total impact), followed by the indicator “human toxicity—non-cancer effects” (35%). Seventy-seven percent of particulate matter emissions and 90% of human toxicity—non-cancer effects are caused by the landfilling of aggregate waste. 

In the economic dimension, for the indicator “LCC and incomes”, 90% of the total cost is for deconstruction workers. There is no income. For the indicator “environmental externalities”, the largest portion of the cost (55%) comes from the landfilling of aggregate waste and the second largest portion (33%) comes from the production and combustion of diesel used for deconstruction. 

#### 3.2.6. Module D

Regarding the benefits and loads beyond the system boundary, the largest potential future benefit in the environmental dimension is the avoidance of contributing to global warming potential (51% of the total benefit) and the avoidance of abiotic depletion for fossil resources (30%), see Figure 5. The main factors involved in the avoidance of contributing to global warming potential are the potential recycling of bridge steel (52%) and the potential re-use of aggregates (45%).

The greatest potential future benefits in the social dimension are the avoidance of particulate matter emissions (61%) and the avoidance of ionising radiation—human health (14%). The main factors involved in the avoidance of particulate matter emissions are the potential re-use of aggregates (59%) and the potential recycling of bridge steel (39%). 

In the economic dimension, for the indicator “LCC and incomes”, 94% of the total income comes from the potential recycling of the bridge’s structural steel plates. For the indicator “environmental externalities”, the largest benefit comes from the potential recycling of the bridge’s structural steel plates (70%).

### 3.3. Comparison of the Design Concepts

A comparison of the two design concepts over the life cycle (modules A–C excluding module B8) and in terms of the future re-use, recovery, and recycling potential (module D) is presented in Table 8 and Figure 7, Figure 8 and Figure 9. The results are aggregated on the dimension level for the environmental and social dimensions and on the indicator level for the economic dimension. The results for the environmental and social dimensions are normalized and weighted, and the results for the economic dimension are summarized. The results for the environmental and social dimensions per life cycle stage for the two concepts are presented in Figure 10, and the summarized results for the economic indicators are presented in Figure 11 (excluding module B8).

In the environmental dimension, the CSF bridge performs better than the SSC bridge over the life cycle (see Table 8 and Figure 7). The environmental impact of the CSF bridge is approximately one-third lower than that of the SSC bridge. After the end-of-life stage (in module D), the potential avoidance of a negative environmental impact is 39% greater for the SSC bridge. 

In the social dimension, the SSC bridge performs better than the CSF bridge over the life cycle (see Table 8 and Figure 8). The social impact of the SSC bridge is 27% lower than that of the CSF bridge. After the end-of-life stage (in module D), the potential avoidance of negative social impact is 22% greater for the SSC bridge. 

In the economic dimension, the SSC bridge performs better than the CSF bridge over the life cycle for the indicator “LCC and incomes”, but it has a significantly worse performance than the CSF bridge for the indicator “environmental externalities” (see Table 8 and Figure 9). The net cost of the SSC bridge is 20% lower than that of the CSF bridge when it comes to the indicator “LCC and incomes”. In contrast, the impact of the factor “environmental externalities” is six times greater for the SSC bridge than for the CSF bridge. After the end-of-life stage (in module D), the potential income and the avoidance of environmental externalities have very small impacts for both concepts. 

Considering the different life cycle stages, the SSC bridge has double the environmental impact in the material production phase (modules A1–A3) compared with the CSF bridge (see Figure 10). This is mainly due to the abiotic depletion potential of non-fossil resources caused by the manufacture of structural steel plates for the SSC bridge. This indicator contributes to almost half of the total environmental impact of the SSC bridge in the production stage (see Figure 7). The indicator global warming potential contributes to one-quarter of the environmental impact of the SSC bridge in the production stage. 

The CSF bridge has a 44% larger social impact in the material production phase than the SSC bridge (see Figure 10). This is primarily due to “human toxicity—cancer effects”, which are mainly caused by production of stainless-steel reinforcement, and the particulate matter emissions, which are mainly caused by the production of aggregates.

In the production stage, the impact of the environmental externalities of the SSC bridge is almost nine times greater than that of the CSF bridge (see Figure 11). This is due to the use of non-renewable elements in the production of the bridge’s structural steel plates. In the construction stage, the net cost for LCC and incomes is almost three times larger for the CSF bridge. This is because of the larger cost for construction workers for the CSF bridge compared with the SSC bridge.

## 4. Discussion

In the environmental dimension, the CSF bridge was found to perform better than the SSC bridge over the life cycle (see Table 8 and Figure 7). The environmental impact of the CSF bridge was 37% lower than that of the SSC bridge. Similar results were demonstrated in an LCA study comparing a steel box girder bridge and a concrete box girder bridge, where the concrete bridge alternative performed best environmentally overall [22]. However, another LCA study on four CSF bridges and four SSC bridges showed the opposite result: the SSC bridges performed better than the CSF bridges over the life cycle [23]. This is partly because only 54–74% of the structural steel mass was used in three of the SSC bridges in the study by [23], compared with the SSC bridge in this case study. It is also partly because 37% of the structural steel plates were assumed to be secondary steel produced in an electric arc furnace (EAF) route in [23], while in this case study, all of the structural steel plates are from primary steel produced through a blast furnace (BF) route.

In the social dimension, the SSC bridge was found to perform better than the CSF bridge over the life cycle (see Table 8 and Figure 8). The social impact of the SSC bridge was 27% lower than that of the CSF bridge. A similar result was demonstrated by [23], where particulate matter emissions were slightly lower for three of the SSC bridges compared with the CSF bridges. 

In the economic dimension, the SSC bridge was found to perform better than the CSF bridge over the life cycle for the indicator LCC and incomes, but it performed significantly worse than the CSF bridge for the indicator environmental externalities (see Table 8 and Figure 9). The opposite result was shown for environmental externalities in [23]. Using the Ecotax02 and Ecovalue08 monetary weighting methods updated with the Ecovalue12 method indicators [24,25], the SSC bridges performed better than the CSF bridges. This might be explained by the fact that the depletion of abiotic resources indicator was not included in the calculation of environmental externalities in [23], even though it is part of both the Ecotax02 and the Ecovalue08 and Ecovalue12 methods. Non-renewable elements and non-renewable energy resources were found to be the major contributors contribute to the environmental externalities in this case study.

For both design concepts, the majority of the negative impact on sustainability was found to occur in the production stage (modules A1–A3). This was also shown in [23], where between 55% and 92% of the environmental impact occurred in the production stage, depending on the indicator. An LCA study of a steel box girder bridge and a concrete box girder bridge similarly showed that the production of materials for the bridge superstructure and the abutments accounted for the main share of the environmental impact, with a limited number of materials being important [22].

Furthermore, the case study demonstrated that 36% of the life-cycle environmental impact for the CSF bridge and 20% for the SSC bridge occurred in the construction stage (modules A4–A5). In [23], it was also shown that the environmental impact of the construction stage is significant; causing up to 34% of the life cycle impact for some indicators. However, in [22], the construction phase accounted for a relatively small part of the impact, and the use phase contributed more significantly, which is contrary to the results of this case study. A main difference between the two concepts in this case study is that a large part of the economic impact (LCC and incomes) was found to occur in the construction stage for the CSF bridge, and this was mainly due to the cost of construction workers. 

Module B8 was found to contribute to 50% and 38% of the total environmental impact and 66% and 73% of the total social impact over the life cycle (modules A–C) for the CSF bridge and SSC bridge, respectively. This demonstrates that the environmental and social impacts of the bridge itself are, in fact, significant in comparison to the impact from traffic on the bridge. This was even more obvious for the economic impacts, as module B8 was shown to only contribute to between 3% and 17% for the two economic indicators considered for both bridge types. 

The main contribution to the environmental impact over the life cycle (modules A–C excluding B8) was shown to come from the indicator abiotic depletion potential for non-fossil resources for the SSC bridge (35%) and the indicator global warming potential for the CSF bridge (44%). Similar results were demonstrated in an LCA study of a concrete box girder bridge and a steel box girder bridge where global warming, abiotic depletion, and acidification were found to be the indicators with the greatest contributions [22]. In [23], on the contrary, it was shown that the SSC bridges performed better than the CSF bridges regarding the indicator global warming potential. This was partly because less structural steel was used in the SSC bridges and because 37% of the structural steel plates were assumed to be secondary steel produced via an EAF route in [23] (see further explanation above). If only the indicator “global warming potential” had been considered in this case study, the SSC bridge would have performed only 7% worse than the CSF bridge in the environmental dimension. When considering all environmental indicators, the SSC bridge was found to perform 60% worse in the environmental dimension. This demonstrates the importance of including more indicators than only global warming potential, as shown in previous studies [26]. This is an important observation, as today, it is common practice to solely consider global warming potential (or one other indicator such as embodied energy) in assessments of environmental performance [9,26,27].

The main contributor to the social impact over the life cycle (modules A–C excluding B8) was shown to be the indicator particulate matter emissions for both concepts. This is, in part, because of the large weight given to this indicator, but also because construction activities are significant sources of particulate matter emissions, for example, when crushing aggregates [28,29]. For the CSF bridge, the indicator “human toxicity—cancer effects” was also shown to contribute to a large portion of the social impact. For the CSF bridge, it was shown that approximately one-third of the particulate matter emissions were caused by the production of aggregates and their transport to the construction site, and one-third were caused by the production and combustion of diesel used at the construction site. For the SSC bridge, it was shown that one-third of the particulate matter emissions were caused by the production of aggregates; one-third by the production and combustion of diesel used for the construction, maintenance, and deconstruction of the bridge over the life cycle; and one-quarter by the production of structural steel. 

The net cost for the LCC and incomes indicator was found to be 25% higher for the CSF bridge than the SSC bridge over the life cycle. For the CSF bridge, the main costs came from cost for workers during construction, and the production and transport of aggregates, concrete, and reinforcement steel. For the SSC bridge, the main contributor to the cost of the LCC and incomes was the cost for the production of structural steel. The environmental externalities of the SSC bridge was six times greater than that of the CSF bridge over the life cycle because of the use of non-renewable elements in the production of the bridge’s structural steel plates. 

The production of aggregates and their transport to the construction site was shown to be the main factor in the environmental and social impacts of the CSF bridge and the social impact of the SSC bridge. It was also shown to be the second greatest factor in the environmental impact of the SSC bridge. Hence, there is great potential to reduce environmental and social impacts by re-using aggregates on site in the next life cycle to avoid the production of virgin aggregates and their transport to the construction site. The production of the bridge’s structural steel plates plays the largest role in the environmental impact as well as the impact on environmental externalities for the SSC bridge due to the depletion of metals. Thus, there is great potential to reduce the impact by using steel produced from recycled steel. Regarding the LCC and incomes over the life cycle, the results show that costs can be reduced by lowering the work costs as well as the material costs for the materials purchased in large quantities, such as aggregates, steel and concrete. 

It is important to keep in mind that the environmental and social impact results are highly dependent on the LCA datasets chosen for the calculations. It is possible to apply the method using generic licensed datasets or generic datasets from open online LCA databases or Environmental Product Declarations (EPDs), provided they follow the EN15804 + A2 standard. The use of supplier EPDs instead of generic datasets further increases the accuracy of the environmental and social assessment results, since EPDs contain supplier-specific declarations, while generic datasets may not be fully representative of the actual materials supplied in an assessed civil engineering works project. In this case study, only generic datasets were used. They are not completely representative of the resources purchased for the object of assessment. For example, the generic dataset used for aggregates differed from the aggregates purchased, especially regarding the distance between the mining site and crushing plant (10 km in the generic dataset, a few hundred metres for the actual supplier). Generic datasets were used because supplier specific EPDs that follow the EN15804 + A2 standard are not yet available. 

As shown from the examination of results, this method allows the sustainability performance of design concepts to be compared at the life cycle stage and construction component level in the early design and planning stages. The data available in these early stages are sufficient for assessment. Through the examination possibilities made available by the transparency of the method, it is possible to identify the critical elements in a civil engineering works project with the greatest impacts on sustainability. This allows necessary adjustments to be made to achieve more sustainable design concepts. Due to its general character, the method can be applied to other types of civil engineering works, not only bridges. 

For the social dimension, in particular, but also for the environmental dimension, further research is needed to define appropriate indicators for civil engineering projects. The sustainability dimensions could also be further aggregated using a multi-criteria decision analysis (MCDA) method to obtain an overall sustainability score [30,31,32]. Furthermore, scenarios for the construction, use, and end-of-life stages may be improved by collecting data from ongoing projects [5]. It is recommended that future studies carry out a sensitivity analysis to assess the influences of different scenarios and datasets whose uncertainty is considered important for the evaluated impacts. 

## 5. Conclusions

The case study demonstrates that our method can be used to carry out comparable and transparent life cycle sustainability performance assessments in the early design and planning stages of a civil engineering works project. It allows the sustainability performance of design concepts to be compared at the life cycle stage and construction component level. The method enables the identification of critical indicators with the greatest impacts on sustainability at the different life cycle stages and for the critical elements. The method is transparent, because the underlying BOMs, scenarios, and datasets used for the assessment are clearly described, and the results can be evaluated down to the building material level. Since the method is based on quantitative indicators and fixed factors, the calculation process used in the assessment is automatable. 

The use of supplier EPDs instead of generic datasets will further increase the accuracy of the environmental and social assessment results, since EPDs are supplier-specific declarations, while generic datasets may not be fully representative of the actual materials supplied in an assessed civil engineering works project.

The case study demonstrates the importance of including more indicators than global warming potential in environmental assessments. Environmental and sustainability performance is clearly dependent on several indicators, and care should be exercised when generalising results obtained in assessments that only take into account only one or a few indicators. 

The method used in the case study includes state-of-the-art indicators according to current standard specifications and can be complemented with additional indicators. For the social dimension, in particular, but also for the environmental dimension, further research is needed to define appropriate indicators for civil engineering projects.

The results of the case study show the importance of the production stage (modules A1–A3) and the construction stage (modules A4–A5) on the sustainability performance over the life cycle. Production of structural steel for the SSC bridge has the greatest environmental impact and accounts for almost all of the environmental externalities, which explains the poorer performance of this bridge in the environmental dimension. However, the CSF bridge was shown to perform worse than the SSC bridge in the social dimension with a higher LCC. The former point can be mainly explained by the large global warming potential of the CSF bridge due to concrete production, and the latter point can be explained by higher costs during construction due to being more labour-intensive. The production and transport of aggregates have large negative environmental and social impacts for both bridge types.

In summary, the examination of the case study assessment results provides important knowledge on the indicators and life cycle stages associated with large sustainability impacts for each of the bridge concepts investigated. We conclude that to reduce the overall negative impact on sustainability, mitigation measures should primarily address the production and construction stages. Our findings contribute to the development of a better understanding of the sustainability impact of civil engineering works through the identification of elements with the greatest impacts. A special focus and adaptations of identified elements (e.g., origin and type of materials, equipment used, structural optimization) could significantly improve the sustainability performance of the design concepts. After necessary adaptations have been applied to a design concept, the assessment can be re-performed to assess the new conditions. If the assessment is automated, this step could be iterated until the most sustainable alternative is found.

## Figures and Tables

**Figure 1 ijerph-17-07909-f001:**
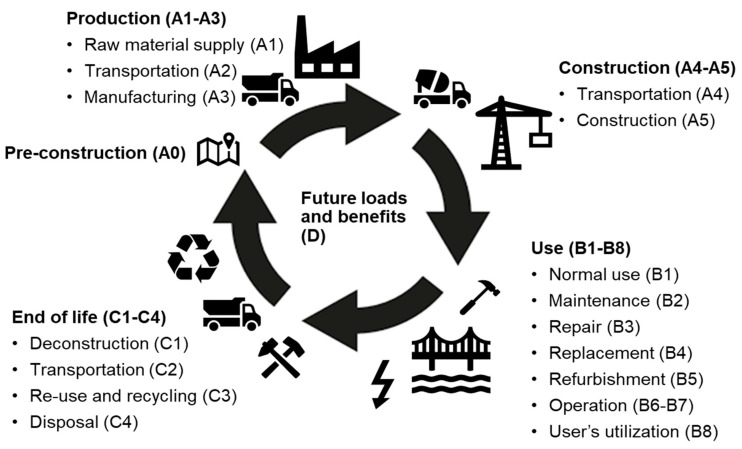
Schematic illustration of the life-cycle stages of a civil engineering works project and their classification in modules.

**Figure 2 ijerph-17-07909-f002:**
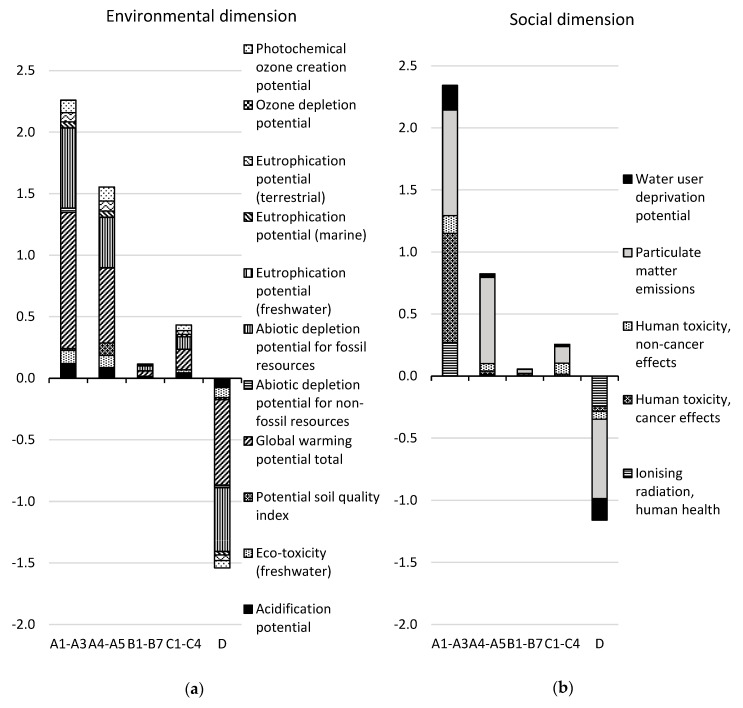
Normalized and weighted results for the CSF bridge design concept per life cycle stage in (**a**) the environmental dimension and (**b**) the social dimension for each indicator and for the functional unit. Module B8 is not included. See Figure 4 for module B8. Note that the ozone depletion potential indicator bar cannot be seen in the figure since it is very small.

**Figure 3 ijerph-17-07909-f003:**
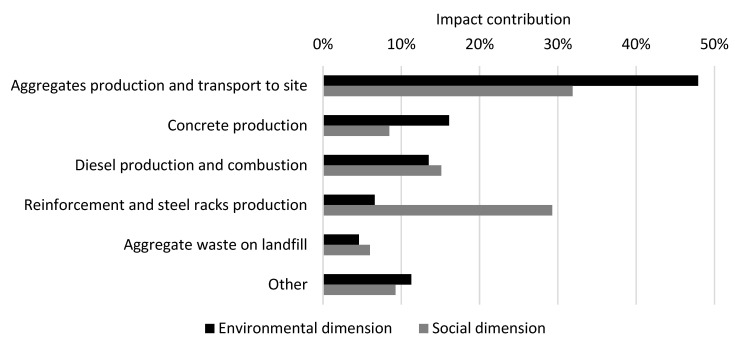
Contribution of resources to the total impact over the life cycle (modules A–C, excluding B8) for the CSF bridge design concept in the environmental and social dimensions.

**Figure 4 ijerph-17-07909-f004:**
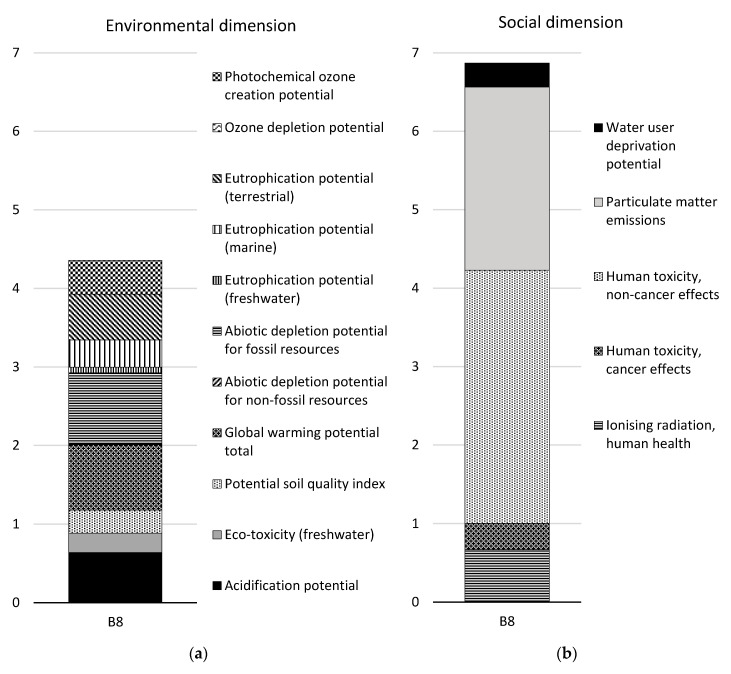
Normalized and weighted results for module B8 for both bridge design concepts in (**a**) the environmental dimension and (**b**) the social dimension for each indicator and for the functional unit. Note that the ozone depletion potential indicator bar cannot be seen in the figure since it is very small.

**Figure 5 ijerph-17-07909-f005:**
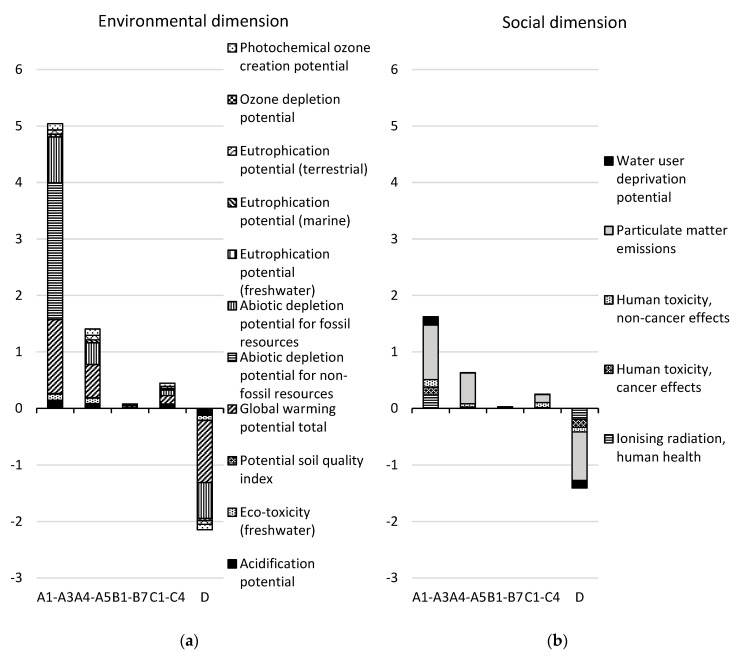
Normalized and weighted results for the SSC bridge design concept per life cycle stage in (**a**) the environmental dimension and (**b**) the social dimension per indicator and for the functional unit. Module B8 is not included. See Figure 4 for module B8. Note that the ozone depletion potential indicator bar cannot be seen in the figure since it is very small.

**Figure 6 ijerph-17-07909-f006:**
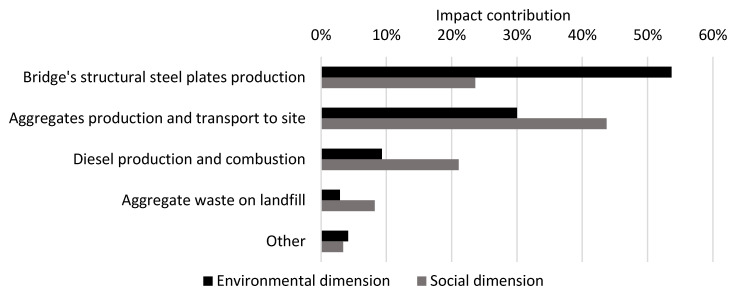
Contribution of resources to the total impact over the life cycle (modules A–C, excluding B8) for the SSC bridge design concept in the environmental and social dimensions.

**Figure 7 ijerph-17-07909-f007:**
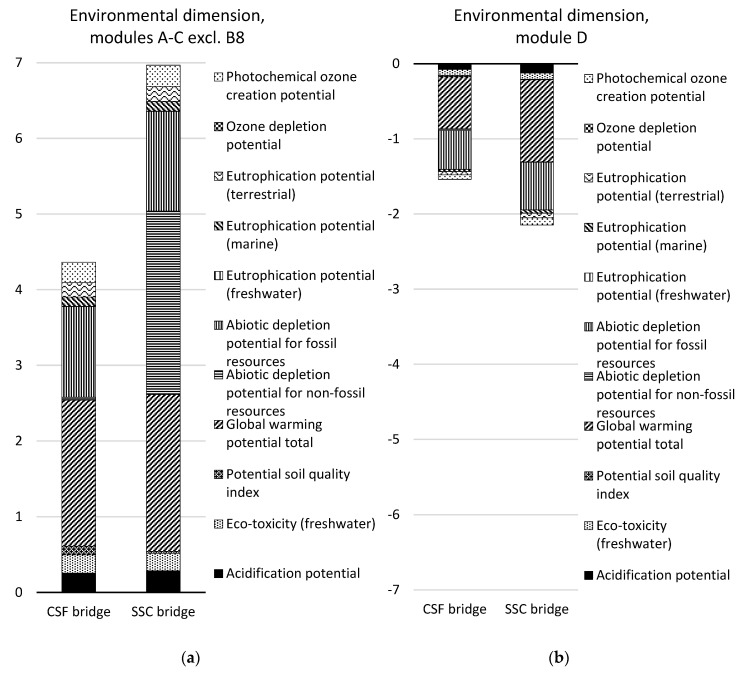
Comparison of the design concepts in the environmental dimension for (**a**) life cycle stages A–C excluding module B8 and (**b**) module D per indicator and for the functional unit. Note that the ozone depletion potential indicator bar cannot be seen in the figure since it is very small.

**Figure 8 ijerph-17-07909-f008:**
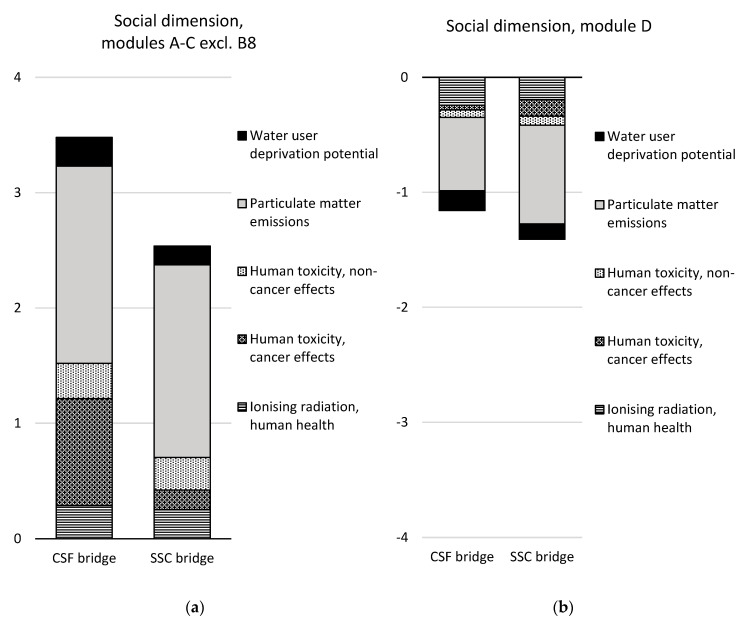
Comparison of the design concepts in the social dimension for (**a**) life cycle stages A–C excluding module B8 and (**b**) module D per indicator and for the functional unit.

**Figure 9 ijerph-17-07909-f009:**
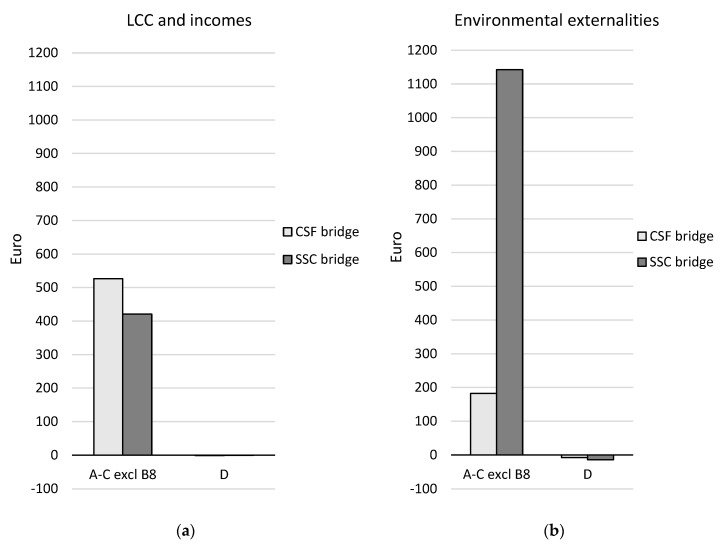
Comparison of the design concepts per life cycle stage for (**a**) the LCC and incomes and (**b**) the environmental externalities; presented in Euros for the functional unit. Module B8 is not included in the comparison.

**Figure 10 ijerph-17-07909-f010:**
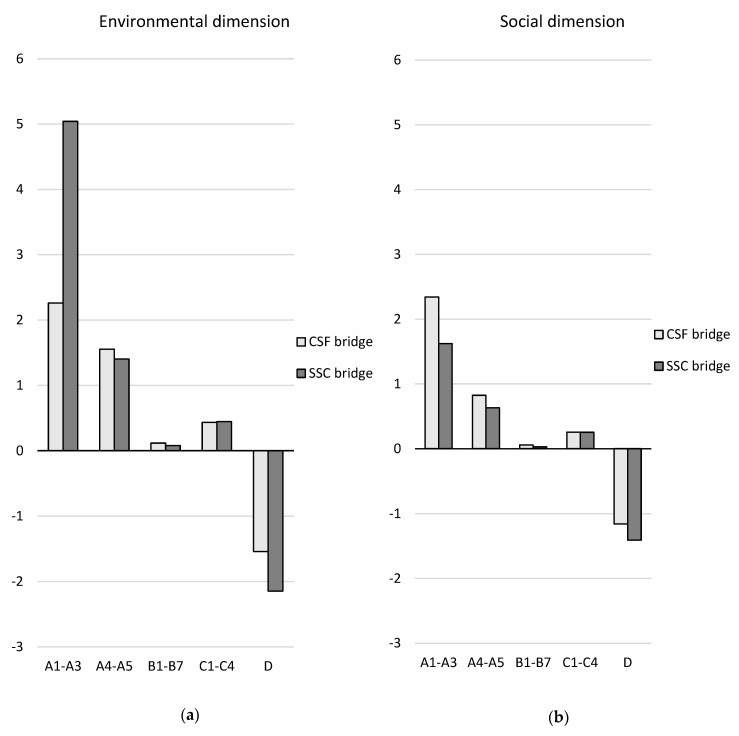
Comparison of the design concepts per life cycle stage in (**a**) the environmental dimension and (**b**) the social dimension for the functional unit. Module B8 is not included in the comparison.

**Figure 11 ijerph-17-07909-f011:**
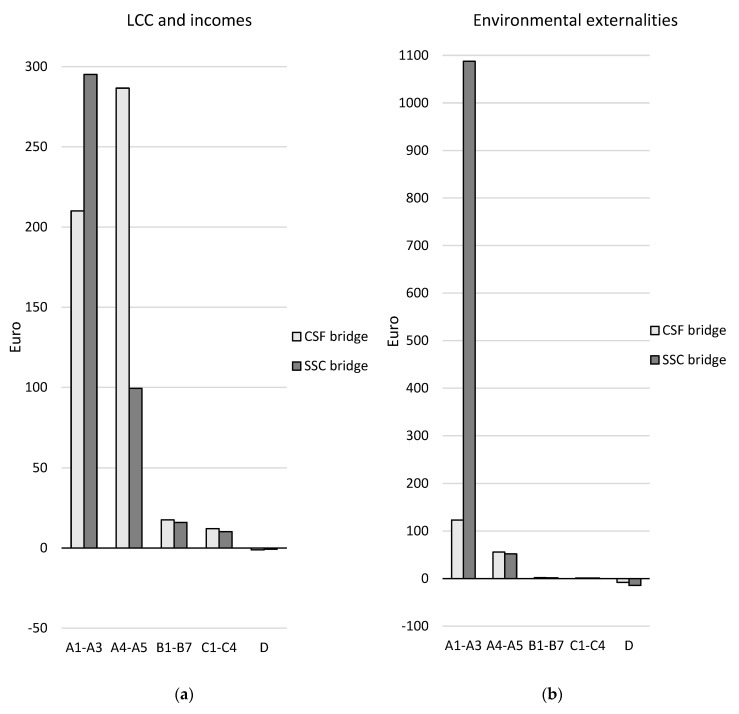
Comparison of the design concepts per life cycle stage for (**a**) the LCC and incomes and for (**b**) the environmental externalities presented in Euros for the functional unit. Module B8 is not included in the comparison. Note the different y-axis scales in the charts.

**Table 1 ijerph-17-07909-t001:** Case study specific prerequisites for the assessment.

Characteristic	Case Study Prerequisite
Object of assessment	Bridge 6-1282-1 on Road 26, Sweden 6 m long, 9 m wide, 3 road lanes
Intended use of the assessment	Design concept comparison
Additional functions provided	-
Functional equivalent:	
(a) Type/use of the civil engineering works, (b) Capacity, (c) Reference study period and pattern of use, (d) Design life (required service life, RSL)	(a) Road bridge with fauna passage (b) 7200 AADT, 100 km/h (c) 80 years, see Appendix C, Appendix D (d) 80 years
Time of assessment in the life cycle	Detailed design/tender phase
Life cycle stages assessed	A1–A5, B1–B8, C1–C4, D
Justification of the exclusion of modules	A0 was excluded because of its insignificant impact on the sustainability performance and because it does not differ between the concepts.
Area of influence	Environmental, social, and economic dimensions (environmental externalities): The surroundings and people in the direct vicinity of the bridge, receiving emissions from fuel combustion and other activities during construction, use and deconstruction from passing vehicles across the length of the bridge. Economic dimension (Life cycle costs (LCC) and incomes): The users of passing vehicles on the bridge (module B8), the client of the constructed bridge (all other modules).
Energy and mass flows considered in the assessment	See Appendix B
General assumptions and scenarios used	See Appendix B
Sources of data for the indicators	See Appendix B
Statement about whether data are specific or generic	See Appendix A
Reference year for the cost data	2019

**Table 2 ijerph-17-07909-t002:** Normalization and weighting factors used for environmental and social indicators [11].

Dimension	Indicator	Normalization Factor (NF)	Weighting Factor (%)
Environmental	Acidification potential	55.6	8.43
	Eco-toxicity potential (freshwater)	42,683	2.61
	Potential soil quality index	819,498	10.80
	Global warming potential total (fossil + biogenic + luluc)	8096	28.63
	Abiotic depletion potential for non-fossil resources	0.0636	10.27
	Abiotic depletion potential for fossil resources	65,004	11.31
	Eutrophication potential (freshwater)	1.61	3.81
	Eutrophication potential (marine)	19.5	4.02
	Eutrophication potential (terrestrial)	177	5.04
	Ozone depletion potential	0.0536	8.58
	Photochemical ozone creation potential	40.6	6.50
Social	Potential ionizing radiation—human health	4220	18.94
	Human toxicity potential—cancer effects	0.0000169	8.05
	Human toxicity potential—non-cancer effects	0.000230	6.96
	Particulate matter emissions	0.000595	33.88
	Water user deprivation potential	11,469	32.17

**Table 3 ijerph-17-07909-t003:** Sustainability dimensions, categories, indicator names, abbreviations, and units of measurement for the indicators.

Dimension	Category	Indicator Name	Abbreviation	Unit of Measurement
Environmental	Acidification	Acidification potential	AP	mol H + eq
	Biodiversity	Eco-toxicity potential (freshwater)	ETP-fw	CTUe
		Potential soil quality index	SQP	Dimensionless
	Climate change	Global warming potential total (fossil + biogenic + luluc)	GWP-total	kg CO_2_ eq
	Depletion of abiotic resources—minerals and metals	Abiotic depletion potential for non-fossil resources	ADPE	kg Sb eq
	Depletion of abiotic resources—fossil fuels	Abiotic depletion potential for fossil resources	ADPF	MJ, net calorific value
	Eutrophication	Eutrophication potential (freshwater)	EP-freshwater	kg P eq
		Eutrophication potential (marine)	EP-marine	kg N eq
		Eutrophication potential (terrestrial)	EP-terrestrial	mol N eq
	Ozone depletion	Ozone depletion potential	ODP	kg CFC 11 eq
	Photochemical ozone creation	Photochemical ozone creation potential	POCP	kg NMVOC eq
Social	Health and comfort	Potential ionizing radiation—human health	PIR	kBq U235 eq
		Human toxicity potential—cancer effects	HTP c	CTUh
		Human toxicity potential—non-cancer effects	HTP nc	CTUh
		Particulate matter emissions	PM	Disease incidence
		Water user deprivation potential	WDP	m^3^ world deprived eq
Economic	Life cycle economic balance	LCC and incomes	-	Euro
	External cost	Environmental externalities	-	Euro

**Table 4 ijerph-17-07909-t004:** Results for the concrete slab frame bridge (CSF) bridge design concept per indicator in units of measurement per life-cycle stage for the functional unit. Modules not assessed are abbreviated as “MNA”.

Indicator	A0	A1–A3	A4–A5	B1–B5	B6–B7	B8	C1–C4	D
AP	MNA	0.78	0.57	0.049	0	4.2	0.29	−0.18
ETP-fw	MNA	1780	1642	133	0	4063	380	−333
SQP	MNA	1002	7588	22	0	22,083	167	−579
GWP-total	MNA	313	172	12	0	234	47	−70
ADPE	MNA	2.1 × 10^−4^	1.6 × 10^−5^	4.4 × 10^−6^	0	5.4 × 10^−5^	3.7 × 10^−6^	−1.0 × 10^−4^
ADPF	MNA	3740	2344	230	0	5229	584	−892
EP-freshwater	MNA	5.5 × 10^−4^	5.6 × 10^−4^	2.6 × 10^−5^	0	3.110^−2^	1.1 × 10^−4^	−2.3 × 10^−4^
EP-marine	MNA	0.24	0.25	0.015	0	1.68	0.10	−0.05
EP-terrestrial	MNA	2.61	2.82	0.17	0	20.4	1.1	−0.5
ODP	MNA	6.9 × 10^−11^	2.6 × 10^−11^	3.1 × 10^−11^	0	4.9 × 10^−10^	8.5 × 10^−14^	−3.5 × 10^−13^
POCP	MNA	0.64	0.71	0.046	0	2.7	0.28	−0.14
PIR	MNA	60	3.1	0.57	0	148.5	0.39	−13
HTP c	MNA	1.8 × 10^−6^	5.6 × 10^−8^	1.2 × 10^−8^	0	7.1 × 10^−7^	2.9 × 10^−8^	−5.8 × 10^−8^
HTP nc	MNA	4.8 × 10^−6^	2.0 × 10^−6^	4.3 × 10^−7^	0	1.1 × 10^−4^	2.9 × 10^−6^	−6.9 × 10^−7^
PM	MNA	1.5 × 10^−5^	1.2 × 10^−5^	5.8 × 10^−7^	0	4.1 × 10^−5^	2.4 × 10^−6^	−3.2 × 10^−6^
WDP	MNA	71	11	0.86	0	109	6.3	−16
LCC and incomes	MNA	210	287	18	0	86	12	−1.1
Environmental externalities	MNA	123	56	2.0	0	31	1.4	−7.8

**Table 5 ijerph-17-07909-t005:** Results for the CSF bridge design concept per life cycle stage aggregated on the dimension level (and on the indicator level for the economic dimension) for the functional unit. The results for the environmental and social dimensions are normalized and weighted, and the results for the economic dimension are summarized. Modules not assessed are abbreviated as “MNA”.

Dimension	Indicator *(Unit)*	A0	A1–A3	A4–A5	B1–B7	B8	C1–C4	D
Environmental	All *(dimensionless)*	MNA	2.3	1.6	0.12	4.4	0.43	−1.5
Social	All *(dimensionless)*	MNA	2.3	0.82	0.057	6.9	0.26	−1.2
Economic	LCC and incomes *(Euro)*	MNA	210	287	18	86	12	−1.1
	Environmental externalities *(Euro)*	MNA	123	56	2.0	31	1.4	−7.8

**Table 6 ijerph-17-07909-t006:** Results for the SSC bridge design concept per life cycle stage and aggregated at the dimension level (or at the indicator level for the economic dimension) for the functional unit. The results for the environmental and social dimensions are normalized and weighted, and the results for the economic indicators are summarized. Modules not assessed are abbreviated as “MNA”.

Dimension	Indicator *(Unit)*	A0	A1–A3	A4–A5	B1–B7	B8	C1–C4	D
Environmental	All *(dimensionless)*	MNA	5.0	1.4	0.078	4.4	0.44	−2.1
Social	All *(dimensionless)*	MNA	1.6	0.63	0.029	6.9	0.25	−1.4
Economic	LCC and incomes *(Euro)*	MNA	295	99	16	86	10	−0.8
	Environmental externalities *(Euro)*	MNA	1 087	52	1.5	31	1.3	−14

**Table 7 ijerph-17-07909-t007:** Results for the SSC bridge design concept per indicator in units of measurement per life-cycle stage for the functional unit. Modules not assessed are abbreviated as “MNA”.

Indicator	A0	A1–A3	A4–A5	B1–B5	B6–B7	B8	C1–C4	D
AP	MNA	0.95	0.56	0.031	0	4.2	0.34	−0.8
ETP-fw	MNA	1773	1571	89	0	4063	360	−1348
SQP	MNA	977	781	13	0	22,083	156	−644
GWP-total	MNA	369	164	9.1	0	233	42	−309
ADPE	MNA	1.5 × 10^−2^	1.4 × 10^−5^	7.5 × 10^−7^	0	5.4 × 10^−5^	3.5 × 10^−6^	−2.9 × 10^−5^
ADPF	MNA	4667	2210	149	0	5222	554	−3646
EP-freshwater	MNA	6.3 × 10^−4^	5.0 × 10^−4^	1.2 × 10^−5^	0	3.1 × 10^−2^	9.8 × 10^−5^	−4.1 × 10^−4^
EP-marine	MNA	0.24	0.25	0.0089	0	1.7	0.12	−0.20
EP-terrestrial	MNA	2.6	2.8	0.10	0	20	1.4	−2.2
ODP	MNA	3.8 × 10^−11^	2.6 × 10^−11^	2.3 × 10^−14^	0	4.9 × 10^−10^	8.5 × 10^−14^	−1.5 × 10^−12^
POCP	MNA	0.69	0.70	0.028	0	2.7	0.36	−0.59
PIR	MNA	53	2.2	0.39	0	149	0.39	−43
HTP c	MNA	2.9 × 10^−7^	3.3 × 10^−8^	9.4 × 10^−9^	0	7.1 × 10^−7^	2.8 × 10^−8^	−3.0 × 10^−7^
HTP nc	MNA	4.4 × 10^−6^	1.9 × 10^−6^	1.4 × 10^−7^	0	1.1 × 10^−4^	2.9 × 10^−6^	−2.6 × 10^−6^
PM	MNA	1.7 × 10^−5^	9.6 × 10^−6^	2.9 × 10^−7^	0	4.1 × 10^−5^	2.5 × 10^−6^	−1.5 × 10^−5^
WDP	MNA	53	1.9	0.47	0	109	2.5	−48
LCC and incomes	MNA	295	99	16	0	86	10	−0.8
Environmental externalities	MNA	1087	52	1.5	0	31	1.3	−14

**Table 8 ijerph-17-07909-t008:** Comparison of the design concepts for modules A–C and for module D for the functional unit. The results are aggregated at the dimension level for the environmental and social dimensions and at the indicator level for the economic dimension. The results for the environmental and social dimensions are normalized and weighted, while the results for the economic dimension are summarized. Module B8 is not included in the comparison. The best options are highlighted in grey.

Dimension, Indicator *(unit)*	CSF Bridge		SSC Bridge	
	A–C	D	A–C	D
Environmental, all *(dimensionless)*	4.4	−1.5	7.0	−2.1
Social, all *(dimensionless)*	3.5	−1.2	2.5	−1.4
Economic, LCC and incomes *(Euro)*	526	−1.1	421	−0.8
Economic, Environmental externalities *(Euro)*	182	−7.9	1 142	−14

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
