# Peer review of "Life Cycle Sustainability Performance Assessment Method for Comparison of Civil Engineering Works Design Concepts: Case Study of a Bridge"

_ijerph, 2020, doi:10.3390/ijerph17217909_

Round 1
Reviewer 1 Report
Very interesting article, which presents a sustainability assessment method that is applied in a road bridge case study. The paper is well structured, and the tables and figures provide a lot of detail and information about what the text wants to describe. The writing of the text is very concise, with simple and accurate writing for a scientific article. The topic is of interest for the scientific community and most especially for practitioners, because the presented method enables to realise life cycle sustainability performance assessments in the early stages of a real civil engineering project. However, there are some points that this reviewer feels need to be improved and completed:
Lines 37-39: “Current standards provide the general framework for the sustainability assessment of civil engineering works but do not give detailed guidance on the calculation of indicators and their aggregation”: I miss a reference at the end of the sentence.
Lines 48-49: “This paper evaluates a method for life cycle sustainability assessment and comparison of civil engineering works design concepts presented in (11) by applying it in a road bridge case study”: improve the statement and improve the way to insert reference 11, it should appear at least the name or names of the authors of the work.
Lines 55-67: I recommend putting this whole paragraph in a schematic way, not everything in a text block; it is not easy to follow.
Line 69: please, improve the way to insert reference 11
Line 98-99: “LCC and incomes as well as environmental externalities are presented as the net present value (NPV) using a discount rate of 3%.” Why has a discount rate of 3% been selected? Please justify and reason it.
Table 2: better describe and reason why these indicators and their weights have been selected.
Lines 115-116: “Scenarios were developed based on expert knowledge and literature data”: better explain and describe this point, describe how the scenario definition process has been developed.
Reviewer 2 Report
This is a very interesting topic, the concept proposed can be further used for general civil engineering activities. Some comments:
- Line 18: since the proposed concept covers the entire life cycle stages, please highlight it in the abstract.
- Line 48: the concept was proposed in the authors' previous study, it not needed to give detailed information in a followed case study paper. However, it is not enough that only point out the aim of this concept, please include more information about this concept, such as components and how it works.
- Line 55, it will be better if the authors can provide a flow chart to explain the entire life cycle stages. It will be easier for readers.
- The analysis in chapter 3.1 and 3.2 are reliable.
- Figure 6: is there 'ozone depletion potential' in the figures? The legend is not easy to distinguish, please revise the filler or add the legend in the figures.
- Figure 7 is blurry, please revise it.
- The conclusion is well summarized.
Reviewer 3 Report
The paper presented evaluates a method for life cycle sustainability assessment and comparison of civil engineering works design concepts presented in another bibliographic document by applying it in a road bridge case study. The study has two primary aims: 1) to analyse the practical application potential of the method and 2) to examine the results obtained in the case study to identify critical indicators in different life cycle stages as well as critical elements in the civil engineering works project with the greatest impacts.
The work is well written and well structured. Also, it is very complete. The cited references are current and abundant. The only thing to improve is the quality of the Figures and their size to try to adjust them to the size of the pages. On the other hand, the Introduction chapter could be extended.
